# The Role of the MeToo Route in Improving the Health of Gender-Based Violence and Isolating Gender Violence Survivors

**DOI:** 10.3390/healthcare12232480

**Published:** 2024-12-09

**Authors:** Paula Cañaveras, Ana Burgués-Freitas, Mar Joanpere

**Affiliations:** 1Department of Sociology, Faculty of Economics and Business, University of Barcelona, 08007 Barcelona, Spain; 2Department of Sociology, Faculty of Political Science and Sociology, University of Granada, 18071 Granada, Spain; anaburgues@ugr.es; 3Department of Business Management, Faculty of Business and Economics, Universitat Rovira i Virgili, 43204 Reus, Spain; mar.joanpere@urv.cat

**Keywords:** isolating gender violence, gender Violence, MeToo, survivors, health improvement, support networks

## Abstract

**Background/Objectives**: The scientific literature has provided evidence on the negative health effects experienced by those who suffer gender-based violence (GBV) and isolating gender violence (IGV), the latter being a form of retaliation against those who support GBV victims. However, less attention has been paid to the potential health improvements following the initial support received by victims. **Methods**: This study examines the positive health outcomes among survivors of GBV and IGV after they engaged with the “MeToo route,” an initiative of the MeToo movement aimed at raising awareness about gender violence and fostering solidarity through support networks that traveled through 13 Spanish universities through more than 20 events in September 2022. **Results**: Using communicative methodology, survivors shared how their health, previously harmed by their experiences of violence, improved as a result of the support provided after knowing the MeToo support network. **Conclusions**: The findings highlight the crucial role of solidarity networks in alleviating the health impacts of GBV and IGV and underscore the importance of effective support systems for recovery.

## 1. Introduction

Gender-based violence, particularly intimate partner and sexual violence, is a major public health issue and a violation of women’s human rights, affecting millions of women and girls globally [1]. Recent estimates indicate that approximately 1 in 3 women (30%) have experienced physical and/or sexual violence from intimate partners or other perpetrators at some point in their lives, totaling around 736 million women. Notably, nearly one-third (27%) of women aged 15–49 who have been in a relationship report experiencing some form of violence from their partner [2]. This violence has severe negative consequences for women’s physical, mental, sexual, and reproductive health, increasing the risk of sexually transmitted infections, including HIV, as reported by the World Health Organization (2024). According to research by Cho and colleagues [3], multiple victimizations, or experiencing different forms of gender-based violence, are associated with chronic pain, headaches, sleep difficulties, and poor health perception [4,5].

Some of the literature has highlighted that gender-based violence is deeply rooted in gender stereotypes and cultural sexist norms, which foster the cognitive distortions that allow this violence to persist [6]. The research line presented here identifies socialization and, specifically, attraction to violence as the root of the issue, proposing that prevention-focused interventions targeting socialization offer an effective solution, a perspective supported by existing studies [7,8]. In Spanish universities, this problem is further compounded by hierarchical, feudal, and entrenched structures that have historically perpetuated the silencing of sexual harassment cases within these institutions [9]. As some research has pointed out, for these approaches to be successful, coordinated collaboration among diverse institutions and organizations working on the ground is essential [10].

In the legal regulation of gender violence in Europe, countries like France have incorporated international concepts into national policies, particularly since the creation of a Ministry for Women’s Rights in 2012 and the ratification of the Istanbul Convention in 2014. However, there remains some resistance to the term “gender” and to adopting a structural approach to gender violence, as reflected in public debates, policies, and attitudes [11].

In Spain, the fifth macro-survey on violence against women conducted in 2015 estimates that 12.5% of women in the country have experienced abuse from a partner at some point in their lives [12]. In this context, the legal framework addressing gender-based violence has evolved significantly over the years. The first law, Ley Orgánica 1/2004, passed on 28 December [13], on comprehensive protection measures against gender-based violence, specifically addressed violence within intimate partner relationships, thereby excluding many women who experienced violence from other sources, such as family members or acquaintances. However, recognizing the limitations of this approach, reforms have been made to broaden the scope of protection. The Ley Orgánica 10/2022 [14], which amends the previous legislation, now includes measures to combat all forms of gender-based violence, thus ensuring that more women receive the necessary support and protection regardless of the context in which the violence occurs. This change marks a significant step towards a more inclusive approach to combating gender-based violence in Spain, acknowledging that the issue affects women in diverse and complex situations.

However, recent ratifications have led to significant advancements by also addressing isolating gender violence (IGV) experienced by those who support victims. The impact of victimization extends beyond those who directly experience gender-based violence; it also affects individuals who stand up for and support victims, often facing retaliation as a result. Isolating gender violence (IGV) refers to the attacks and reprisals directed at supporters of gender violence victims, as perpetrators aim to sever these individuals from their sources of support [15]. This form of violence, previously referred to as second-order sexual harassment, has been redefined through recent dialogic research that amplifies victims’ voices, advocating for its recognition as isolating gender violence [16]. This concept is already being incorporated into legislation by some parliaments, as was the case in the Catalan Parliament in December 2021 [17]. 

In a recent study by Flecha and colleagues [18], survivors of IGV reported a range of physical and mental health issues, including insomnia, gastrointestinal disorders, physical fatigue, arrhythmias or altered heart rates, migraines, loss of appetite or disordered eating, dizziness, eczema or herpes outbreaks, weight loss, and, though less common, muscle pain and allergic reactions. Other studies have also examined the negative health effects on victims of gender-based violence (GBV) when those who stand up to protect them experience isolating gender violence (IGV). This creates a double process of revictimization, which victims perceive as even more painful than the violence they themselves endured. The IGV experienced by those supporting the victims causes the victims to relive some of the health effects from their own victimization, such as persistent anxiety and nervousness [19].

### 1.1. The Impact of Social Support on the Health Improvement of Gender and Sexual Violence Survivors

The scientific literature highlights that social support has a significant positive impact on the physical and mental health of survivors of gender-based and sexual violence. Research conducted in Belgium with asylum seekers who had survived sexual and gender-based violence showed improvements in both physical and mental health after receiving support [20]. Many of these survivors, who exhibited symptoms of PTSD (post-traumatic stress disorder) and psychosomatic conditions due to the violence, experienced a reduction in its effects through peer and family support, which provided both emotional and material assistance.

In another case involving male refugees who had survived sexual violence, mutual support and collective action were identified as crucial factors in their healing process. Their recovery began as a result of peer support and the creation of solidarity networks that helped them overcome their trauma [21]. Positive effects have also been observed in men who support survivors of gender-based violence [15]. These men, who often suffer isolating gender violence (IGV) as retaliation for their stance against violence, experience greater health protection when they feel strongly supported by a solid and reliable network.

Social support can be critical for the transition from victim to survivor. Without adequate support, many victims of violence do not report their situation, which perpetuates their suffering and limits their chances of recovery [22,23]. Evidence suggests that social support, in various forms (peers, family, and community networks, among others), significantly improves the physical and mental health of survivors of sexual and gender-based violence [24].

### 1.2. The MeToo Journey: Amplifying Voices Across Universities

In this context, various movements have emerged, such as the MeToo University, which began as a support network for victims of sexual harassment simultaneously in the United States and Spain. The network MeToo Universidad in Spain is composed of survivors of sexual harassment and isolating gender violence as well as supporters within Spanish universities. This network includes university students, professors, and other members of the academic community. It was initially formed by individuals who, since the 1990s, had taken a stand against sexual harassment in universities [25]. As a result of their efforts, they faced isolating gender violence for daring to expose a widespread problem that had, until then, remained largely unspoken within universities [26].

In the context of academia, the #MeToo movement has revealed concerning statistics regarding gender-based violence and sexual harassment. A 2018 study in the UK found that 41% of 1839 students experienced sexualized behavior from staff. In Ireland, a survey of over 6000 students revealed that 43.6% had suffered unwanted sexual touching, including completed or attempted penetration. Additionally, a survey of 3516 staff members in Ireland showed that 11% of women and 7% of men reported having received unwanted sexual attention, while 10% experienced unwelcome sexual comments online [27].

There is scientific literature that highlights the multifaceted nature of the debate surrounding the #MeToo movement, emphasizing the importance of recognizing not only its successes but also the criticisms and challenges it has encountered. Some studies suggest that the movement has been co-opted or silenced in various contexts, drawing attention to aspects of reality that are frequently overlooked [28,29]. Nevertheless, this research aims to concentrate on the positive impacts of MeToo as a network of solidarity and support, as well as its potential benefits for the health and well-being of victims.

As Joanpere and colleagues [25] explain, this solidarity network achieved significant successes, including the requirement for all universities to have protocols in place for addressing sexual harassment and gender-based violence, the legislation on isolating gender violence, and the growing number of individuals who, feeling supported by a movement that backs victims and their allies, began to share their experiences as victims as a first step toward improving their situations. 

In this context, in September 2022, coinciding with the start of the academic year, the MeToo route was organized. During the week of 26–30 September, two routes, the northern and the southern, traveled across 13 Spanish universities to raise awareness of their strong stance in support of survivors, serving as a network of solidarity. This was achieved through both formal events such as conferences and classroom sessions, as well as informal activities, totaling 20 events. 

The planning of this initiative involved sending invitations to a wide range of Spanish universities, with the aim of covering as much territory as possible through the northern and southern routes. While many universities responded quickly, others joined as the route progressed, despite invitations being sent well in advance. Some universities either declined or did not respond.

The southern route spanned five days, covering the cities of Valencia, Almería, Granada, Málaga, Sevilla, Córdoba, and Extremadura, while the northern route took four days to visit Barcelona, Zaragoza, Deusto, Donosti, Santander, and Oviedo [30]. Numerous people from the MeToo movement participated in the preparation of the route, although only eight people physically participated in the routes, including victims of sexual harassment at university who explained their cases and how it was only thanks to the support they had received that they had managed to become survivors. Each route consisted of a small team travelling by car, which ensured flexibility and close coordination along the way.

The initiative was funded through a crowdfunding campaign on GoFundMe, which raised a total of EUR 2419.50. These funds were allocated to cover the platform’s commission, car rentals, fuel, accommodation, per diems, parking fees, and other transport-related expenses for the eight participants who carried out the route. Contributions came from both individuals and groups, with supportive messages from men and women. A detailed breakdown of the funds raised and their allocation is available on the WorldMeToo website: https://www.worldmetoouniversities.net/ (accessed on 1 December 2024).

The project operated without industry sponsorship or formal institutional backing. It was organized and executed by members of the MeToo network, who were responsible for its planning and implementation. During that time and in the subsequent months, the MeToo Universidad solidarity network continued receiving an increasing number of calls from victims seeking support to overcome their situations. Today, it remains active, offering guidance and assistance to those who reach out. 

## 2. Materials and Methods

The research presented here builds on previous research analyzing the MeToo University network and aims to specifically analyze the impact of the MeToo route carried out in 2022, as a small sample of the full impact it has had and continues to have. The authors of the present article are part of the solidarity network MeToo University in Spain.

### 2.1. Design

This article combines two data collection techniques: (1) a questionnaire, which generates a higher volume of responses from participants shortly after the MeToo events, primarily focusing on assessing the overall impact of those events; and (2) interviews conducted two years later with a selection of participants in order to detect the long-term consequences of having participated in the events carried out by the MeToo movement. While the questionnaire specifically aims to gather information regarding the recognition and identification of sexual harassment at the university and the understanding of the actions taken, the interviews are designed to provide deeper insights into the effects of sexual harassment or isolating gender violence on the health and recovery of alleged victims and participants, two years after the MeToo route.

The interviews were conducted using the communicative methodology (CM) [31], a methodology based on the principle of cultural intelligence and the communicative capacity of all people, recognized by the European Commission for its social impact and ability to drive change [32]. Furthermore, this methodology, due to its focus on co-creation with research participants, has been considered particularly suitable when working with vulnerable populations, such as ethnic minorities and violence survivors [33].

Through this methodology, all research participants engage in an ongoing egalitarian dialogue with the researchers, where participants share knowledge from their personal experiences and researchers contribute insights based on scientific evidence. Through this exchange of reflections, perspectives, and arguments, new knowledge is co-created. This process of egalitarian dialogue or co-creation is essential for the research to generate social impact, meaning it contributes to improvements in society and addresses issues that have been democratically identified as important for progress. The goal of CM is not merely to describe reality but to enhance it through dialogue with communities and individuals [34,35]. These are the initial hypotheses to which the research combining the two techniques aims to provide an answer:

**H1.** 
*Access to the “MeToo route” support network significantly improves the emotional well-being of survivors of gender-based violence and isolating gender violence.*


**H2.** 
*Survivors of gender-based violence who participate in the “MeToo route” experience a reduction in physical symptoms associated with stress and anxiety.*


**H3.** 
*Survivors of gender-based violence and isolating gender violence who access support networks such as the “MeToo route” report a reduced sense of social isolation.*


**H4.** 
*The presence of a clear institutional positioning and commitment to support victims of gender-based violence contributes positively to the emotional health and sense of safety of survivors in the university context.*


**H5.** 
*Awareness-raising generated through MeToo network events motivates participants to get involved in protecting and supporting victims of gender-based violence.*


### 2.2. Data Collection

The two data collection techniques used for this article are differentiated below. After the MeToo route concluded, key contacts involved in organizing the events across universities and other venues were contacted. A questionnaire was distributed through these contacts to reach participants via social media, virtual campuses (in the case of students), email, or other suitable channels. The attendees were invited to complete a form to voluntarily share their reflections on the experience and its impact in the weeks following the event. 

The questionnaire was hosted on the University of Granada’s Google Drive platform, and consisted of only three questions related to participant profiles to ensure privacy, along with four questions addressing the impact generated by the event. The information collected is as follows:City where you attended the event.Type of event in which you have taken part (possible answers: (1) official event, (2) talk given in a subject class (in the classroom), (3) informal social event (e.g., in a restaurant, jazz club, or ‘tapas’ bar).Profile of the person (education, professional, or academic background).What topics have you reflected on after attending the event? (open-ended response).Do you think attending the event could help you identify cases of sexual harassment at the university? (yes/no).Has the event provided you with tools on how to act in the event of detecting gender-based violence or harassment? (yes/no).If you would like to add any further comments, feel free to write whatever you wish (open-ended response).

A total of 111 participants completed the form, providing responses to both closed-ended questions, yielding data for percentage analysis, and open-ended questions, offering a free space for expression that was subsequently analyzed using qualitative techniques.

Of those who participated in the survey, 57.7% were university students, and 20.7% were university professors. Among the other participants, 12.6% were non-university teachers, and there were also individuals holding leadership positions within the university, as well as members of the general public. Of these participants, 87.4% attended the official events organized during the route, 18% took part in social gatherings with the route members (such as dinners and meetings in cafés), and 9.9% attended talks held in university classes along the route (see Table 1).

Secondly, two years after the conclusion of the MeToo route, the organizers of the events at the different universities were contacted again to ask for contact details of people who would like to conduct an in-depth interview. Four in-depth interviews were conducted with specific profiles described below to gain further insights (see Table 2). These interviews provide valuable long-term perspectives on the outcomes. 

The individuals selected for these interviews contributed by sharing personal examples of how their health had improved as a result of participating in the #MeToo route. The consent form explained that participation in the study was voluntary, that no personal information would be disclosed, and that participants could withdraw at any time. All participants signed a written informed consent, agreeing to take part in the study and to be audio-recorded.

For both the questionnaire and interview participant selection, organizers were asked to assist in reaching out to attendees from diverse backgrounds, including university professors, students, and staff, regardless of specific roles or affiliations. Organizers were encouraged to connect with individuals who had expressed interest in the event and were willing to participate in the research, thereby ensuring a varied group of voices and experiences. Those who agreed to take part in the study were subsequently invited to share their personal opinions and experiences regarding the impact of the MeToo network.

All participants in the study remain anonymous, and their data have been handled in accordance with the consent they provided. Their privacy and confidentiality were strictly maintained throughout the process, ensuring that their identities are protected.

### 2.3. Data Analysis

To perform the data analysis for the questionnaire, all responses were compiled in an Excel document to enable calculation of percentages and generation of graphs. Only responses in the “Other” section of Question 3, which asked participants to “define your profile,” were recoded to ensure more clarity and relevance in the responses. In the analysis, the percentages for Questions 5 and 6, which addressed the talk’s direct impact on providing tools for action against gender-based violence, were of particular importance. Open-ended Questions 4 and 7, designed as reflective spaces, were analyzed using the same thematic categories as those applied to the in-depth interviews.

The interviews were transcribed and thoroughly reviewed multiple times by the research team to facilitate in-depth data analysis. Using the communicative methodology, categories were developed by identifying inclusive and exclusionary elements (see Table 3). Specifically, elements were categorized based on whether they contribute positively or negatively to participants’ physical and mental health (considering aspects such as emotional well-being, feelings of social isolation, stress- and anxiety-related physical symptoms, and sense of safety). Additionally, elements were identified based on their role in fostering participants’ awareness of the issue and encouraging proactive engagement.

Through repeated reading of the interview transcripts and responses to the open-ended questions in the questionnaire, other relevant categories emerged, such as the role of institutional positioning. In a collaborative dialogue, the researchers established four key categories for analysis:Negative physical and/or mental health symptoms as a consequence of suffering from GV or IGV.Symptom amelioration.Importance of institutional positioning and action against bullying at universities.Awareness-raising and what motivates participants to get involved in protecting and supporting victims.

## 3. Results

Overall, regarding the data obtained from the four interviewed participants, survivors reported significant improvements in their physical and mental health, which had previously been severely impacted by their experiences of violence in the form of sexual harassment or isolating gender violence. Many described struggling with symptoms such as stress, isolation, or psychological distress before receiving support. After learning about the MeToo support network through their route through the different universities, participants reported feeling calmer and experiencing an improvement in symptoms related to the violence they had endured. This progress was largely attributed to the sense of safety and connection they felt within the MeToo support network.

Additionally, regarding the 111 questionnaire participants, 91.7% of the responses indicated that the event helped them identify cases of sexual harassment at the university, while 5.5% responded that it maybe helped and 2.8% stated that it did not. Similarly, 82.2% reported that the MeToo event provided them with tools on how to act in cases of gender-based violence or harassment, with 16.2% indicating that it may have provided such tools and 3.6% stating that it did not.

The results are presented as follows: first, the negative health consequences of sexual harassment and isolating gender violence are outlined; next, the improvements in health attributed to the sense of support from a network are discussed; and finally, the importance of institutional public positioning in creating safer university environments is examined.

### 3.1. Negative Health Symptoms Resulting from Gender-Based Violence and Isolating Gender Violence

The first result highlights the negative impact that experiencing sexual harassment in the university or being subjected to isolating gender violence can have on the mental and physical health of survivors. Three of the four participants from the interviews acknowledged having experienced SH or IGV and its consequent health effects, while the fourth did not experience violence herself but witnessed a female classmate and friend go through it during her university years. Specifically, as articulated by a participant, the health of a university worker who at the time was a predoctoral student, Iris, deteriorated when, despite many being aware of the violence, no one took action, leaving her to carry the burden. Some of the symptoms she describes include stress, a sense of isolation, and burnout. Iris noted that “that did psychological damage because I felt alone,” indicating that despite holding a position of greater authority, the lack of institutional support left her feeling vulnerable and overwhelmed. Moreover, Iris’s account underscores the co-occurrence of different types of harassment and their compounded effects on health. Ultimately, this overwhelming situation and its effects led her to abandon her academic career at that time within the university.
*So, yes, that did psychological damage because I felt alone. Since I had more power than the students, I felt obliged to take action, but it’s true that you don’t always have the mental strength to do everything and pursue a career. If there were more transparency, people would understand the weight of getting involved, and you’d feel more supported. People would be more aware, but no one took a stance. (…) I definitely experienced burnout. (…) In my case, the consequence was abandoning my doctoral thesis, and that was a very negative factor. (…) The person before me, another research fellow, developed bipolar disorder under the same supervision (…) due to overexertion and not stopping when serious issues arose.*
*I helped with addressing the harassment of students, but then I also experienced workplace harassment. So, these two things coexisted in space and time for me. When it comes to sexual harassment towards students, it was very important that there was a movement because, in my case, it was something I could study, a phenomenon where different types of harassment coexist at various levels within universities. That has a massive impact on health; on one hand, you see that students, for example, in my case, because I was a research fellow at the time, are being harassed. You see that no one reacts. Only someone connected to the Women’s Institute, and I (so, a full professor and myself) took action. It’s very stressful. And for me, it did affect my health a lot because it’s something that requires a lot of work since there are no measures in place.*(Iris)

The personal testimonies of survivors provide critical insight into the direct health consequences of gender-based violence (GBV) and isolating gender violence (IGV). In the following excerpt, Pearl reflects on the physical and mental health consequences experienced as a result of gender-based violence. She describes how the prolonged stress and anxiety caused by the violence led to various health conditions, such as irritable bowel syndrome and fibromyalgia, which were diagnosed at a young age. Pearl’s testimony highlights the impact that violence has had on her well-being, illustrating the deep connection between gender-based violence and chronic health issues.
*Yes, I mean, for example, in my particular case, I can say it, because I believe there have been situations of gender-based violence and I have that syndrome, I mean, I have irritable bowel syndrome. (…) Because of that, it affects my stomach. And not only my stomach, but also many times… (…) So, of course, in my case, it also affected me in terms of fibromyalgia, and I was diagnosed very young because of the stress and anxiety that the violence caused me. So, yes, I do see it as very important (…).*(Pearl)

Finally, Amelia’s testimony sheds light on the significant emotional and academic consequences her friend experienced as a result of harassment by a professor during their university years. Although Amelia does not provide a detailed account of the specific health effects, she highlights that her friend suffered from these health consequences. The fear of encountering this professor not only created a persistent sense of anxiety but also had wider repercussions on academic performance and personal well-being. Amelia mentions how the harassment affected both her friend and another classmate, leading to negative impacts such as lowered grades and strained interactions within the academic environment. Furthermore, the situation took a toll on her friend’s mental health and professional aspirations, as pursuing an academic career became intertwined with the constant fear of facing the abuser. The lack of institutional support compounded these effects, as Amelia notes that even professors in other departments were aware of the situation but offered no assistance.
*In the end, my colleague didn’t… no. She wanted to dedicate herself to the University and, in fact, later on, she pursued it. Well, she is pursuing an academic path. But she was, of course, afraid of encountering this professor in the same University (…) We ended up having him again at another university. There were repercussions in grades and everything, towards her and towards the group we sat with, as well as towards another classmate who had to take another course with him due to a medical issue and had him again. And this classmate also suffered consequences in terms of grades, treatment, everything. And, I mean, the one who suffered the most also had a clear idea that she wanted to do research. And although she pursued it, it did affect her, not only in her health because, well, going with the fear of running into him does take a toll, but also in the opportunities she had. She realized that, even after choosing professors from other departments, they also knew that this professor did those things and she had no support. So, that also had repercussions. And then, when I went to the MeToo event, I really thought a lot about this.*(Amelia)

### 3.2. Improvement of Health Symptoms Due to Support Received

Iris’s reflection highlights the transformative power of support and institutional acknowledgment in the aftermath of experiencing gender-based violence. Her insights reveal how the emergence of the MeToo movement fostered a sense of coordination and collective awareness among survivors and advocates. By witnessing the proactive measures taken in Valencia, she felt reassured that systemic changes were being implemented, which contributed to her emotional healing and overall well-being. Iris articulates how this newfound awareness and institutional commitment allowed her to shift her focus back to her work, alleviating the burden of unresolved tension. Her experience underscores the critical importance of supportive environments in mitigating the negative health impacts of past trauma and in promoting a safer, more equitable future for all individuals.
*About MeToo, I think, yes, it would have been important if there had been a MeToo there [She is referring to the case she experienced at her university before the MeToo passed through]. When the news broke about the professor who was harassing, everyone became aware and protested, but (…) Seeing it in Valencia helped me a lot, like, it made me feel like there was coordination. It really does, I mean… It relieved me to think that awareness was being raised, like it was being implemented in institutions and that there was an agenda with follow-up. That really gave me a lot of peace and calm (…) And like, a sense of reparation, you know, the idea that not all places are the same. And it’s so closely tied to health, I don’t know how to explain it; when you feel calm and you feel at peace, you say, well, okay. Now I can focus on my work, you know, I don’t have that unresolved tension that I have to carry alone, or whatever. (…) Now there’s a path where you can improve society and improve reality so that it doesn’t happen again to other people, and that really lowers your stress levels a lot.*(Iris)

Just as participants described how their physical health changed before and after hosting the route at their universities, the survey conducted at the end of the route showed that, out of 111 respondents, 69 reported a change in their mindset due to the initiative. These participants expressed that the route helped them to become more aware of the issues surrounding sexual harassment and gender-based violence. In particular, it encouraged them to start thinking about how they could care for others who might be suffering from these forms of abuse or their consequences. One participant, in particular, highlighted the importance of meeting survivors who have found a sense of well-being, as it showed that it is possible to transform this situation, including through personal recovery. This not only provided hope, but also served as an example of how individuals can overcome these challenges, inspiring others to contribute to the healing and support process.

### 3.3. The Importance of Institutional Stance and Action Against Harassment for Victims’ Sense of Security in Universities

In this regard, participants emphasized the role that institutional support and positioning played in supporting victims by embracing the MeToo movement. This institutional backing not only validated their experiences but also provided a sense of hope and reassurance essential for their emotional well-being. Knowing that institutions are willing to acknowledge the issue of sexual harassment can significantly alleviate the burden on victims, who often feel isolated and powerless. Participants felt that when those in power publicly acknowledge the importance of addressing sexual harassment, it legitimizes their experiences and promotes a supportive environment. As they mention, this recognition not only contributes to a safer atmosphere for victims but also positively impacts their mental health, as it reduces feelings of isolation and increases the likelihood of seeking help.
*Yes, well, it’s true that any action, any visibility measure already gives you hope, like a sense that something can be done, and that calms you inside. Because otherwise, you feel like the decision for absolutely everything falls on you, and that you don’t have the strength to match the power position of the accused. (…) I was surprised that there were representatives from the institution [on the MeToo event she attended]. (…) I really appreciated that representatives from the university came. I don’t know if it was from the faculty (…) the representative at that time, and I liked it a lot. She explained and said outright that this event was very important to them, that the topic was critical, and that they were going to incorporate it into their agendas. I thought, well, it’s great that the awareness piece happened, because it wasn’t just about people understanding what sexual harassment is and that it still happens today, not something that happened in the past, etc., but also that people with different power positions within the university were present, and that there was a verbal consensus about the importance of this issue and that it was going to be included in their agendas. For me, that’s already a huge step, because in my experience at the previous university, in the Faculty of Education, when there were cases, not just suspicions of sexual harassment (…).*(Iris)

In the following quote, Melissa reflects on her experience upon arriving at the University of Extremadura, coinciding with the emergence of the MeToo movement in the academic environment. Despite her brief stay at the university, she highlights how her transition from an environment marked by gender violence and emotional pressure to a more supportive atmosphere was transformative. She describes this change as liberating, emphasizing the significance of the support she received and the tangible difference in the atmosphere of the place. Melissa also notes how her new experience has impacted not only her well-being but also those who participated in that space, suggesting that institutional support can be crucial for healing the wounds of the past.
*I have only been at the University of Extremadura for a very short time. I arrived in March and this was in September, and I had just come from a process of, first going through the University of XXXX, with what happened there in terms of gender violence and the people around us suffering, and then to another university where this type of pressure was suffered a lot. It is true that this feeling of stress and powerlessness was felt. And of feeling very small. And to arrive in Extremadura was to give another opportunity, and to coincide with the MeToo movement was very special. I think it was liberating in that sense. That’s the word to define the tidal wave of feelings at that moment. Because of the support received (…) it was all very easy. (…) The response was immediate. (…) The atmosphere was so different from what I had brought with me, it was a break with what existed before. (…) it was an earthquake, and it has had an impact on the people who participated in that space. (…) Some time later I remember having some emails in which they thanked me for having that space to be able to address those issues.*(Melissa)

Pearl also reflected on how the institutional changes, especially in terms of leadership and equality measures, that were made at her university since the MeToo event had a direct impact on her sense of safety and support within the university environment. The feeling of having a reliable support system, especially from key institutional figures, gave them a sense of confidence and protection in dealing with any gender-related challenges:
*Yes, it’s true that having a support network has really helped me. For example, the changes that have happened in the dean’s office regarding equality and such have made me feel safer and more supported. If I ever need anything from the team in charge, I’ve felt quite backed up.*(Pearl)

In the following quote, Amelia reflects on how her perception of the university environment transformed after attending the MeToo event. Initially, she viewed the university as a hostile space due to hearing about cases of violence. However, hearing success stories of survivors who overcame challenges and now thrive within supportive networks gave her a renewed sense of hope. These stories shifted her view, showing her that the university could also be a place of healing and positive change, providing spaces free from violence and offering crucial support systems.
*It gave me a sense that those spaces within the University could be nice, because I had heard (…) throughout my studies, cases like those mentioned in the MeToo events. And I had somewhat of a vision of the University as a very hostile space, full of violence. It never crossed my mind that it could be a positive environment. So, I remember it as hearing very tough things, but also very beautiful because everything shared were success stories. So, listening to how that transformation happened and how, despite all the difficulties they had faced, they managed to overcome those situations. And moreover, it’s especially beautiful to see people who are now doing well, even doing very well within the University, and have a strong support network. So, the possibility of being in these spaces opens a future to places without violence. (…) What it conveyed to me was a sense of great hope and reassurance of “wow, how great that this exists, if only we had known (…) at least to know where to go or where you can find support.*(Amelia)

Similarly, this result was underlined by many of the questionnaire respondents who highlighted, when asked about key issues after attending the event, the importance of raising awareness about violence and establishing strong support for victims of gender-based violence and isolating gender violence. They made it clear that universities should serve as safe spaces for victims and their allies, not for perpetrators, highlighting the need for a clear position from both institutions and the wider community on this issue.

One student, reflecting on the Granada event in the questionnaire, noted “*I have thought about the importance of raising awareness of violence within the university, how to act when we suspect something is happening to our peers and the importance of supporting victims*”. This comment sums up the urgency felt by many participants to establish a supportive framework that ensures the safety of victims and, at the same time, empowers their peers to take effective action.

Similarly, a non-university faculty member present at the Valencia event noted that “*We must work across the board to take a courageous stand against violence and harassment, always supporting the victims and those who may suffer isolating gender violence*”. This statement highlights the collective responsibility of the academic community to foster an atmosphere of solidarity and support while advocating proactive measures to address the prevalence of harassment. Another non-university faculty participant at the Valencia event also expressed that “*It has been crucial to amplify diverse voices and listen to the success story of [one of the survivors sharing her story on the road], as it provided us with keys to fight together against sexual violence at the university and beyond. I thought that what is clear is that the ones who should be out of the universities are the bullies and that we cannot leave anyone alone, neither the victims nor the supporters*”.

Finally, a university faculty member present at the Santander event noted “*The positioning of an institution is the first essential step in making the community aware that sexual harassment exists at the university and that measures will be put in place to prevent it*”. This idea sums up the essence of this subsection: that institutional commitment is not just a procedural necessity, but a fundamental step in fostering a safer academic environment and the only possible one in which students can fully persevere in their academic goals.

## 4. Discussion

Based on the findings, the hypotheses are justified as follows: H1 is fully supported, as survivors reported significant improvements in emotional well-being due to the support provided by the ‘MeToo route’ network, which offered a sense of safety and emotional relief. H2 is partially supported, as some survivors noted improvements in physical symptoms associated with stress and anxiety, such as irritable bowel syndrome and fibromyalgia; however, this was not universal, as some focused more on emotional relief than physical relief. H3 is fully supported, with testimonies highlighting that accessing this support network reduced feelings of social isolation, as survivors found a safe and supportive space within the university. H4 is also fully supported, as the clear institutional commitment to combat GBV had a positive impact on survivors’ sense of safety and emotional health. Finally, H5 is fully supported, as the awareness-raising efforts of the ‘MeToo route’ events motivated participants to become more involved in supporting victims, as demonstrated by their increased commitment to solidarity and advocacy against violence.

The negative effects of experiencing violence, both physical and mental, are well documented [4,19]. Survivors often suffer from conditions such as post-traumatic stress disorder (PTSD), chronic stress, anxiety, depression, and a range of physical symptoms, including gastrointestinal issues or chronic pain, such as fibromyalgia in the case of one of the participants. These symptoms can significantly impair a person’s quality of life, as highlighted in various accounts, demonstrating the deep and lasting impact that gender-based violence (GBV) and sexual harassment can have on one’s health [5]. However, research and personal testimonies from participants in this research suggest that receiving support and having a reliable network can greatly improve these outcomes [23]. When survivors feel supported, especially through institutional backing or peer solidarity networks, the negative effects on their health can be mitigated or even reversed, leading to enhanced well-being [24].

Through participants’ accounts, the role of institutions in this context cannot be understated. When institutions openly position themselves against violence, take responsibility for supporting victims, and commit to protecting their interests, survivors report experiencing a sense of security and trust [23]. This institutional accountability is crucial because it offers survivors the possibility of a safer environment where, in the event of violence, there is likely to be a response that prioritizes their well-being. In this way, the transparency and accountability shown by institutions can help to increase victims’ confidence [18].

Thus, the MeToo movement on its journey through universities in the majority of regions in Spain can serve as a powerful example of how institutions can build on this responsibility. By embracing the MeToo initiative, universities are showing their commitment to making campuses safer spaces with the potential to provide not only a greater sense of relief and safety to survivors but also by sending a clear message to those who perpetrate or condone violence that the institution is no longer indifferent. One of the significant achievements of this movement, in line with the scientific literature, is that universities and solidarity networks have collaborated closely, acknowledging that each plays a distinct and essential role in the fight against gender-based violence [10].

According to participants, movements such as MeToo and other solidarity networks that implement effective measures can thus achieve more than just raising awareness, even reducing and/or preventing cases of sexual harassment, gender-based violence and isolating gender violence (IGV). By establishing accountability and support structures, these movements can serve as preventive health measures. By fostering environments where violence is not tolerated and ensuring that support for survivors is easily accessible, these initiatives can actively reduce the severe health impacts associated with violence.

## 5. Conclusions

The MeToo route in Spain transformed the context, not only in addressing sexual harassment in universities but also in improving the health and well-being of those who participated in the events, engaged in discussions, and took action against the issue. In the narratives analyzed and the voices gathered during the journey, many girls who attended the events had the opportunity to change and break their silence, with some even sharing personal experiences they had faced. The evidence collected here demonstrates how actions that shift the perspective from victims to survivors and encourage breaking the silence can create a space for public health. This is particularly relevant when considering the development of public actions and policies aimed at addressing these issues.

Finally, it is important to note that these interviews were conducted two years after the events, allowing the study to capture long-term consequences. This timing is deemed relevant, as it allows for the observation of sustained health and emotional improvements reported by participants over an extended period. It is uncommon to observe long-term effects resulting from a single event, which suggests that the MeToo initiative may have had a profound and enduring impact on the well-being of survivors. This emphasizes the initiative’s potential to promote lasting positive outcomes beyond the immediate aftermath of the events. Although the lack of representativeness is acknowledged, the results still demonstrate how initiatives like the MeToo create safe spaces that facilitate the transition from victims to survivors, improving health outcomes and also serving a preventive function. 

### Limitations and Future Lines of Research

One limitation of this study is that, despite the large number of participants engaged through the questionnaires and in-depth interviews at various MeToo events, the sample cannot be considered fully representative, as the results only reflect the experiences of those participants who chose to respond. We acknowledge that this self-selection may introduce a degree of bias, as we lack information on non-respondents. Nevertheless, the responses we gathered through the in-depth interviews are deemed particularly meaningful as they provide significant value by offering rich qualitative data that enhances the depth of the research findings from individuals who attended the MeToo events. 

Our methodology did not employ a control group; instead, it focused on capturing participants’ perceived changes in themselves following these events. Through the communicative methodology, which emphasizes egalitarian dialogue and co-creation of knowledge with participants, we aimed to ensure that the research process was both inclusive and impactful. This method allowed participants to contribute insights from their own experiences, while researchers provided scientific context, leading to the co-creation of knowledge. By employing this approach, we sought findings that are not only descriptive but also socially impactful.

Future research could focus on expanding the sample size and diversity to achieve greater representativeness, including participants from different cultural, academic, and social backgrounds present at the events. Additionally, longitudinal studies would be valuable to examine the long-term health impacts of engaging in support networks like those fostered by the MeToo movement. Another avenue for exploration is investigating the specific mechanisms within these initiatives that contribute to survivors’ recovery and resilience, which could inform the development of more targeted interventions.

Building on these findings, practical applications for universities and institutions emerge, centering on the development of scientifically supported support networks that enhance both health and resilience among survivors. Evidence from this study underscores the importance of structured support systems that not only provide immediate assistance but also foster long-term recovery and a sense of empowerment. Policies could focus on establishing clear protocols for addressing gender-based violence, including institutional commitments to action, ongoing education for staff and students, and dedicated resources for survivor support. Additionally, universities can prioritize creating safe spaces where survivors feel supported in sharing experiences, which contributes to collective healing and preventive awareness. By embedding these practices within institutional policy, universities can more effectively support survivor health and well-being, ensuring that initiatives like the MeToo route continue to promote positive, enduring impacts on campus communities.

## Figures and Tables

**Table 1 healthcare-12-02480-t001:** Profile of questionnaire participants.

City	Type of Event	Profile
Barcelona (7.2%) Bilbao (14.4%) Cáceres (13.5%) Granada (26.1%) Madrid (1.8%) Oviedo (8.1%) San Sebastián (1.8%) Santander (5.4%) Sevilla (0.9%) Valencia (17.1%)	Official event (87.4%) Talk given in a classroom (9.9%) Social event (restaurant, jazz club, or tapas bar) (18%)	Student (57%) University professors (20.7%) Non-university teaching staff (12.6%) Others (9.7%) including Federation of Families, NGO technical staff, predoctoral researchers, university administrative and services staff, those in management positions, interested individuals, volunteers, and retired professors

**Table 2 healthcare-12-02480-t002:** Participants’ profiles.

No.	Pseudonym	Profile
1	Iris	Female. Attended the MeToo event in Valencia. Professor profile but a predoctoral researcher at the time of the incidents. She experienced IGV as a result of supporting victims’ cases at her university. (30–40 years old).
2	Pearl	Female. Victim of violence. Attended the event in Granada. Student profile. (20–30 years old).
3	Amelia	Female. During her university years, a friend experienced sexual harassment by a professor. Attended the event in Valencia. Student profile, but now a predoctoral researcher. (20–30 years old).
4	Melissa	Female. Professor who organized one of the MeToo events. She experienced IGV as a result of supporting victims’ cases at her university. (40–50 years old).

**Table 3 healthcare-12-02480-t003:** Categories of analysis and their dimensions.

	Health Consequences	Awareness-Raising and Involvement in the Protection and Support of Victims
Inclusive factors		
Exclusionary factors		

## Data Availability

The data presented in this study are available on request from the corresponding author. The data are not publicly available due to privacy issues and anonymity and of the participants.

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
