# Peer review of "The Role of the MeToo Route in Improving the Health of Gender-Based Violence and Isolating Gender Violence Survivors"

_healthcare, 2024, doi:10.3390/healthcare12232480_

Round 1
Reviewer 1 Report
Comments and Suggestions for Authors
The article is well written and clear to read. The topic addressed is interesting and full of ideas for future further studies and practical applications.
However, I advise the authors to make some changes to improve the paper:
Introduction:
- The authors could clarify the term "Isolating Gender Violence" more directly. It might be beneficial to define it sooner in the introduction to help readers understand its importance from the outset.
- When addressing the achievements of the MeToo movement in academic settings, providing quantifiable data could enhance the discussion (for instance, by referencing reports that include statistical information).
Method:
Provide more information on the type of sampling used.
Results:
In the method the authors say "This approach combined both quantitative data from the form responses and richer, qualitative insights from the interviews to provide a comprehensive understanding of the participants' experiences and the effects the initiative had on their well-being.". But then in the results they do not discuss quantitative data, but only qualitative. Clarify this aspect.
Discussions:
To contextualize what you say, authors could expand on previous studies that have addressed the issue of the consequence of violence against women in Europe (e.g.: Fernández, M. D., Silva, I. M. M., Vázquez-Portomeñe, F., & Calvo, M. S. R. (2017). Features and consequences of gender violence: Study of cases confirmed by a conviction. Spanish journal of legal medicine, 43(3), 115-122.; Di Napoli, I., Procentese, F., Carnevale, S. , Esposito, C., & Arcidiacono, C. (2019). Ending intimate partner violence (IPV) and locating men at stake: An ecological approach. International journal of environmental research and public health, 16(9), 1652.; Allwood, G. (2016). Gender-based violence against women in contemporary France: domestic violence and forced marriage policy since the Istanbul Convention. Modern & Contemporary France, 24(4), 377-394.
Conclusions:
In addition to recommendations for future research, the authors could offer practical applications or policy recommendations that emerge from your findings, which could guide institutions in implementing changes.
Author Response
Thank you very much for your comments and observations, which have greatly enriched our work. We deeply appreciate the time and attention you have dedicated to reviewing our manuscript. Your suggestions and recommendations have been invaluable and have allowed us to enhance the quality and clarity of our article. We are confident that, thanks to your detailed reviews, the manuscript has significantly improved. We sincerely thank you for your effort and dedication.
Comment 1. Introduction: - The authors could clarify the term "Isolating Gender Violence" more directly. It might be beneficial to define it sooner in the introduction to help readers understand its importance from the outset.
- Response 1. Thank you for your valuable feedback. We appreciate your suggestion regarding the clarification of the term "Isolating Gender Violence." In our revised manuscript, we have included additional context about Spanish laws pertaining to gender violence, which we believe enhances the understanding of the term. While "Isolating Gender Violence" was initially introduced in the introduction, we now provide more comprehensive information that highlights its significance and relevance within the legal framework. We hope this clarification will help readers grasp its importance from the outset.
Comment 2. Introduction (2): - When addressing the achievements of the MeToo movement in academic settings, providing quantifiable data could enhance the discussion (for instance, by referencing reports that include statistical information).
- Response 2. We have now incorporated relevant statistics from recent studies, which highlight the prevalence of gender-based violence and harassment in higher education.
Comment 3. Method: Provide more information on the type of sampling used.
- Response 3. Thank you for your valuable feedback on the methodology section. We appreciate your comments, which have guided us in making significant improvements. We have clarified the sampling methods used, provided a more detailed demographic breakdown of participants, and specified our participant selection criteria. Additionally, we have enhanced the discussion around the effectiveness and validity of our chosen methodologies, as well as articulated our study's objectives and initial hypotheses more clearly. We believe these revisions have greatly improved the clarity and rigor of our methodology.
Comment 4. Results: In the method the authors say "This approach combined both quantitative data from the form responses and richer, qualitative insights from the interviews to provide a comprehensive understanding of the participants' experiences and the effects the initiative had on their well-being.". But then in the results they do not discuss quantitative data, but only qualitative. Clarify this aspect.
- Response 4. We have addressed this concern by ensuring that the results now include a more detailed discussion of the questionnaire data alongside the qualitative insights. This enhancement aims to provide a more comprehensive understanding of the participants' experiences and the overall effects of the initiative on their well-being.
Comment 5. Discussions: To contextualize what you say, authors could expand on previous studies that have addressed the issue of the consequence of violence against women in Europe (e.g.: Fernández, M. D., Silva, I. M. M., Vázquez-Portomeñe, F., & Calvo, M. S. R. (2017). Features and consequences of gender violence: Study of cases confirmed by a conviction. Spanish journal of legal medicine, 43(3), 115-122.; Di Napoli, I., Procentese, F., Carnevale, S. , Esposito, C., & Arcidiacono, C. (2019). Ending intimate partner violence (IPV) and locating men at stake: An ecological approach. International journal of environmental research and public health, 16(9), 1652.; Allwood, G. (2016). Gender-based violence against women in contemporary France: domestic violence and forced marriage policy since the Istanbul Convention. Modern & Contemporary France, 24(4), 377-394.
- Response 5. Thank you very much for your valuable feedback. We have taken your suggestion into account and incorporated the recommended references into the article.
Comment 6. Conclusions: In addition to recommendations for future research, the authors could offer practical applications or policy recommendations that emerge from your findings, which could guide institutions in implementing changes.
- Response 6. We have now included a section in the end of Conclusions outlining practical applications and policy recommendations derived from our findings. These recommendations emphasize the importance of establishing effective support networks and implementing scientifically-backed actions to foster safer and more supportive environments within institutions.
Reviewer 2 Report
Comments and Suggestions for Authors
The idea of your article is interesting, my recommendations are the following:
Abstract
Gender-based violence, that is, sexist gender stereotypes, it is important to make visible that we are talking about women. However, this information is missing in the abstract.
Introduction
Due to the current confusion, it would be important to reflect that violence based on gender stereotypes, gender-based violence, responds to cultural sexist stereotypes that make men believe they have rights over women. In fact, there is scientific literature that shows that sexist cognitive distortions are the basis and maintenance of this type of violence.
Likewise, men do not experience gender violence. This is an erroneous categorization (paragraphs 67 et seq.). Furthermore, in Spain Law 1/2004 states that gender violence is limited to men to women, therefore it would be a conception that ranges from illegal to erroneous. Gender violence since Beijing (ONU, 1994) is limited from men to women due to sex, and through gender. Mixing these categories represents a setback and a technical and legal contradiction. Men experience family or domestic violence, not gender violence.
The Mee Too movement was born as a movement to denounce women because by experiencing discrimination based on sex, they are silenced by the sexist social configuration. Not exposing that it is a phenomenon that happens to women, but talking about individuals, means making them invisible and erasing them.
Methods
It would be important to expose a methodological contrast, such as the effectiveness or validity of the measures. Even at a qualitative level there are valid methodologies, different types of methodologies, etc. that are important to expose to see the scientific quality of the work.
It would be important to specifically the sex of the participants: how many are men and how many are women. Likewise, other variables of interest would be interesting such as average age, interval, etc.
Likewise, it would be appropriate to expose the participant selection system, for example, why these 4 profiles of the 11 participants, as well as the impact or rate of generalization of the results.
It would be interesting to present the objectives or starting hypotheses, since that would facilitate what was intended to be measured, and in relation to this, what was found.
Results
Regarding the results, even though they are qualitative, it is important to reflect the methodology on which the approaches are based. For example, the observed patterns, the communalities, the contrast of variables, etc. And that is not recorded.
It would be interesting to reflect whether the objectives or hypotheses were confirmed or not. There is no driving axis that represents unified work.
Discussion
The discussion would lack sources of contrast from those who do not present the same thing. There is literature that reflects how that movement was swallowed up, instrumentalized or tried to silence. Only one side of reality is exposed, and at a scientific level it is important to reflect all the edges to be objective.
Likewise, it would be necessary to cite to clarify what is said about the majority of regions in Spain, since it is not supported by sources.
Conclusion
A disconnection is observed between the results and the conclusion. Precisely because it does not have a guiding axis with objectives, or a systematic pattern of contrasted variables.
References
It would be important to expand with sources that replicate what is stated, as well as consider the highest ranking laws in Spain such as 1/2004, which is not cited and is the one with the highest legal hierarchy. Above the regional ones that are mentioned.
Author Response
Thank you very much for your comments and observations, which have greatly enriched our work. We deeply appreciate the time and attention you have dedicated to reviewing our manuscript. Your suggestions and recommendations have been invaluable and have allowed us to enhance the quality and clarity of our article. We are confident that, thanks to your detailed reviews, the manuscript has significantly improved. We sincerely thank you for your effort and dedication.
Comment 1. Abstract: Gender-based violence, that is, sexist gender stereotypes, it is important to make visible that we are talking about women. However, this information is missing in the abstract.
- Response 1. Thank you for your insightful comment. We completely agree with your observations regarding gender-based violence (GBV) and its specific impact on women. However, we would like to clarify that our study does not focus exclusively on women. As we have now explained in greater depth based on your feedback, the research includes male participants as well—specifically in the questionnaire phase, though not in the in-depth interviews. While we acknowledge that men cannot be victims of GBV, as defined by Spanish and international frameworks, we do include men in the study to highlight how they can experience isolating gender violence (IGV) when they take a stand to support victims of GBV and face retaliation as a result.
Comment 2. Introduction: Due to the current confusion, it would be important to reflect that violence based on gender stereotypes, gender-based violence, responds to cultural sexist stereotypes that make men believe they have rights over women. In fact, there is scientific literature that shows that sexist cognitive distortions are the basis and maintenance of this type of violence.
- Response 2: Thank you for your suggestion. We have incorporated this perspective into the article, emphasizing that violence rooted in gender stereotypes is closely linked to sexist cultural norms. We also highlight evidence that socialization and a propensity for violence are key factors contributing to this issue, particularly in the context of Spanish universities, which have historically operated within feudal structures that have suppressed reports of sexual harassment (see lines 42 and following).
Comment 3. Introduction (2): Likewise, men do not experience gender violence. This is an erroneous categorization (paragraphs 67 et seq.). Furthermore, in Spain Law 1/2004 states that gender violence is limited to men to women, therefore it would be a conception that ranges from illegal to erroneous. Gender violence since Beijing (ONU, 1994) is limited from men to women due to sex, and through gender. Mixing these categories represents a setback and a technical and legal contradiction. Men experience family or domestic violence, not gender violence.
The Mee Too movement was born as a movement to denounce women because by experiencing discrimination based on sex, they are silenced by the sexist social configuration. Not exposing that it is a phenomenon that happens to women, but talking about individuals, means making them invisible and erasing them.
- Response 3: Thank you for your comment; we fully agree with your statement. In the indicated paragraphs (now 104 and following), we specifically refer to men who may be victims of sexual violence, but we do not suggest that these cases involve gender-based violence. We agree that gender-based violence, as defined by Spain’s Law 1/2004 and various international frameworks (now also referenced in the article), refers to violence perpetrated by men against women on the basis of gender.
Our approach includes the idea that men can play an important role in addressing gender-based violence by raising awareness and protecting victims. However, we reiterate that we do not suggest that men can be victims of gender-based violence. Rather, they may experience other forms of violence, such as sexual violence or sexual harassment, which, although it predominantly affects women, can also affect men.
Additionally, when we discuss Isolating Gender Violence (IGV), we consider that anyone can be affected by it, as men who take a stance to protect victims of gender-based violence may also suffer retaliatory attacks or hostility from aggressors or from environments that oppose efforts to combat this form of violence.
Thank you again for your review and constructive comments, which allow us to reinforce and clarify the precise terminology in our work.
Comment 4. Methods: It would be important to expose a methodological contrast, such as the effectiveness or validity of the measures. Even at a qualitative level there are valid methodologies, different types of methodologies, etc. that are important to expose to see the scientific quality of the work.
- Response 4: Thank you for your suggestion regarding the need to clarify the methodological contrast, as well as the effectiveness and validity of our chosen methods. In response, we have reorganized the methodology section to provide a clearer explanation of our approach and its relevance to this study.
The selected methodology, while not representative in the statistical sense, is validated by the European Commission (EC) for its social impact and suitability in research involving vulnerable groups, such as survivors of violence. This means that the findings are specific to respondents who participated in the study, and we acknowledge that we do not have data for those who chose not to respond. Nonetheless, we consider the insights gathered particularly meaningful for those participants who provided responses, as these reflect their unique experiences and perspectives on the impact of MeToo-related events. We have now added this information in the article as well.
Unlike studies that rely on control groups, our methodology does not incorporate a control group, as we are not comparing responses across different conditions. Instead, we focus on capturing participants' self-reported changes in perception and awareness after engaging with MeToo events. This approach, involving questionnaires and in-depth interviews, is recognized as valuable for studies aiming to document perceived personal growth or shifts in awareness, especially in contexts dealing with sensitive issues like this case.
CM also emphasizes egalitarian dialogue, where participants' lived experiences inform the research in partnership with researchers' evidence-based insights. This co-creation process fosters a safe, inclusive environment that respects the voices of all participants and is crucial for generating social impact.
We greatly appreciate your insights and will consider incorporating different methodologies in future research to further enhance the scientific quality of our work. Thus, we believe our methodology is well-suited to address the sensitive nature of our research objectives and is in line with best practices for studies that aim to empower and create meaningful impact among vulnerable groups.
(See also the new section on limitations referring to this subject)
Comment 5. Methods (2): It would be important to specifically the sex of the participants: how many are men and how many are women. Likewise, other variables of interest would be interesting such as average age, interval, etc.
Likewise, it would be appropriate to expose the participant selection system, for example, why these 4 profiles of the 11 participants, as well as the impact or rate of generalization of the results.
- Response 5: Thank you for your valuable feedback on the methodology section. We appreciate your comments, which have guided us in making significant improvements. We have clarified the sampling methods used, provided a more detailed demographic breakdown of participants, and specified our participant selection criteria. Additionally, we have enhanced the discussion around the effectiveness and validity of our chosen methodologies, as well as articulated our study's objectives and initial hypotheses more clearly. We believe these revisions have greatly improved the clarity and rigor of our methodology.
Comment 6. Methods (3): It would be interesting to present the objectives or starting hypotheses, since that would facilitate what was intended to be measured, and in relation to this, what was found.
- Response 6: We greatly appreciate your comment and thank you for the suggestion. We have included the starting hypotheses in the Methods section to clarify the purpose of the measurements conducted and their relationship to the findings.
Comment 7. Results: Regarding the results, even though they are qualitative, it is important to reflect the methodology on which the approaches are based. For example, the observed patterns, the communalities, the contrast of variables, etc. And that is not recorded.
- Response 7: We believe that the changes we have made to the methodology now provide greater clarity on how our findings are grounded. By better reflecting the methodologies employed, including the observed patterns, communalities, and the contrast of categories of analysis.
Comment 8. Results (2): It would be interesting to reflect whether the objectives or hypotheses were confirmed or not. There is no driving axis that represents unified work.
- Response 8: Thank you for your valuable feedback. Based on your suggestions, we have now included the hypotheses, which we then address and contrast in the Results section to provide a clearer, unified framework for the study.
Comment 9. Discussion: The discussion would lack sources of contrast from those who do not present the same thing. There is literature that reflects how that movement was swallowed up, instrumentalized or tried to silence. Only one side of reality is exposed, and at a scientific level it is important to reflect all the edges to be objective.
Likewise, it would be necessary to cite to clarify what is said about the majority of regions in Spain, since it is not supported by sources.
- Response 9: Thank you very much for your valuable comments. In response, we have also incorporated this perspective on the other side of the me too from some studies (lines 135 and following). We have also added that during the organization of these events, several universities in Spain were contacted; however, some either did not respond or presented challenges in facilitating the sessions (lines 164-165). Despite these obstacles, we successfully reached 13 universities, achieving unprecedented reach within the Spanish context, and we hope that more institutions will join in the future
Finally, since the study is not quantitative or representative, we acknowledge the limitation regarding coverage across regions. The data gathered does not represent all regions of Spain, as it is based solely on the responses of individuals who attended the events. This clarification reflects the scope of the study and addresses the lack of a fully representative sample across Spain.
Comment 10. Conclusion: A disconnection is observed between the results and the conclusion. Precisely because it does not have a guiding axis with objectives, or a systematic pattern of contrasted variables.
- Response 10: We have addressed your comment by incorporating the hypotheses and providing a justification for how each one is fulfilled in the Discussion section. This addition establishes a clear connection between our findings and the conclusions drawn, ensuring that the text now has a guiding axis with defined objectives and systematically contrasted variables
Comment 11. References: It would be important to expand with sources that replicate what is stated, as well as consider the highest ranking laws in Spain such as 1/2004, which is not cited and is the one with the highest legal hierarchy. Above the regional ones that are mentioned.
- Response 11: We appreciate your suggestion to expand our sources to include those that replicate our findings and to highlight the significance of higher-ranking laws in Spain, such as Law 1/2004, which we had not previously cited. We have now incorporated relevant citations, including this important legislation, as well as Law 15/2022